# Nucleolin-Sle A Glycoforms as E-Selectin Ligands and Potentially Targetable Biomarkers at the Cell Surface of Gastric Cancer Cells

**DOI:** 10.3390/cancers12040861

**Published:** 2020-04-02

**Authors:** Elisabete Fernandes, Rui Freitas, Dylan Ferreira, Janine Soares, Rita Azevedo, Cristiana Gaiteiro, Andreia Peixoto, Sara Oliveira, Sofia Cotton, Marta Relvas-Santos, Luis Pedro Afonso, Carlos Palmeira, Maria José Oliveira, Rita Ferreira, André M. N. Silva, Lúcio Lara Santos, José Alexandre Ferreira

**Affiliations:** 1Experimental Pathology and Therapeutics Group, Portuguese Institute of Oncology, 4200-162 Porto, Portugal; elisabete.fernandes@ipoporto.min-saude.pt (E.F.); ruifreitas@ua.pt (R.F.); dylan.ferreira@hotmail.com (D.F.); soaresjanine7@gmail.com (J.S.); rita.pereir@hotmail.com (R.A.); cristiana.milhazes.gaiteiro@ipoporto.min-saude.pt (C.G.); andreia.peixoto@ipoporto.min-saude.pt (A.P.); sara.simoes.oliveira@gmail.com (S.O.); sofia.ribeiro.cotton@ipoporto.min-saude.pt (S.C.); marta_frds@hotmail.com (M.R.-S.); luis.afonso@ipoporto.min-saude.pt (L.P.A.); carlospalmeira@ipoporto.min-saude.pt (C.P.); llarasantos@gmail.com (L.L.S.); 2Institute of Biomedical Sciences Abel Salazar (ICBAS), University of Porto, 4050-013 Porto, Portugal; 3Institute for Research and Innovation in Health (i3S), University of Porto, 4200-135 Porto, Portugal; mariajo@ineb.up.pt; 4Institute for Biomedical Engineering (INEB), Porto, Portugal, 4200-135 Porto, Portugal; 5Digestive Cancer Research Group, 1495-161 Algés, Portugal; 6REQUIMTE-LAQV, Department of Chemistry, University of Aveiro, 3810-193 Aveiro, Portugal; ritaferreira@ua.pt; 7REQUIMTE-LAQV, Department of Chemistry and Biochemistry, Faculty of Sciences of the University of Porto, 4169-007 Porto, Portugal; andre.silva@fc.up.pt; 8Pathology Department, Portuguese Institute of Oncology of Porto, 4200-162 Porto, Portugal; 9Immunology Department, Portuguese Institute of Oncology of Porto, 4200-162 Porto, Portugal; 10Health Science Faculty, University of Fernando Pessoa, 4249-004 Porto, Portugal; 11Department of Surgical Oncology, Portuguese Institute of Oncology of Porto, 4200-162 Porto, Portugal; 12Department, Porto Comprehensive Cancer Centre (P.ccc), 4200-162 Porto, Portugal

**Keywords:** gastric cancer, glycomics, glycoproteomics, cancer biomarkers, Sialyl-Lewis A, nucleolin, glycosylation, cancer neoantigens, prognosis biomarkers, precision oncology

## Abstract

Background: Gastric cancer (GC) is a major health burden worldwide, with half of patients developing metastases within 5 years after treatment, urging novel biomarkers for diagnosis and efficient therapeutic targeting. Sialyl-Lewis A (SLeA), a terminal glycoepitope of glycoproteins and glycolipids, offers tremendous potential towards this objective. It is rarely expressed in healthy tissues and blood cells, while it is present in highly metastatic cell lines and metastases. SLeA is also involved in E-selectin mediated metastasis, making it an ideal target to control disease dissemination. Methods and Results: To improve cancer specificity, we have explored the SLeA-glycoproteome of six GC cell models, with emphasis on glycoproteins showing affinity for E-selectin. A novel bioinformatics-assisted algorithm identified nucleolin (NCL), a nuclear protein, as a potential targetable biomarker potentially involved in metastasis. Several immunoassays, including Western blot and in situ proximity ligation reinforced the existence of cell surface NCL-SLeA glycoforms in GC. The NCL-SLeA glycophenotype was associated with decreased survival and was not reflected in relevant healthy tissues. Conclusions: NCL-SLeA is a biomarker of poor prognosis in GC holding potential for precise cancer targeting. This is the first report describing SLeA in preferentially nuclear protein, setting a new paradigm for cancer biomarkers discovery and targeted therapies.

## 1. Introduction

Gastric cancer (GC) is the third leading cause of cancer-related death worldwide and constitutes a significant therapeutic hurdle due to the lack of efficient therapeutics to address disseminated disease [1,2]. As such, half of patients develop metastases within 5 years after curative treatment comprehending surgery, chemotherapy and/or radiotherapy [3,4]. Moreover, many patients cannot tolerate or complete systemic treatment due to severe treatment side-effects [5,6]. The introduction of antibody-based targeted therapeutics with trastuzumab (anti-HER2), ramucirumab (anti-VEGFR2), and cetuximab (anti-EGFR) has provided only modest improvements in patient survival and frequently fail to control metastatic spread [7,8,9,10]. In addition, these options serve a reduced number of patient subpopulations and lack the necessary tumor specificity to fully avoid off-target related toxicity [11]. These clinical challenges center the spotlight on a novel quest for cancer-specific molecular targets capable of overcoming significant inter- and intra-tumor heterogeneity.

Malignant transformation is accompanied by a profound remodeling of the plasma membrane glycome and glycoproteome, generating cancer unique molecular fingerprints for precise cancer targeting [12,13]. Premature stop in O-glycan (in Serine and Threonine residues) extension in membrane bound proteins as well as oversialylation and fucosylation of distinct glycoepitopes are among the most well-known events [13,14]. Namely, gastric tumors often overexpress sialylated Lewis (SLe) antigens, which are terminal epitopes of glycan chains in proteins and lipids present at the cell surface [15]. In addition, it has been noted that gastric tumors have a significant increase in the expression of the sialyl Lewis A isoform (SLeA), showing only modest increases of sialyl Lewis X (SLeX) levels [16]. It is also well documented that these sialylated Lewis antigens mediate binding to E-selectin, promoting tumor cells arrest in blood vessels, extravasation into the bloodstream and homing to distant organs [12,17,18]. In fact, the interaction between SLe-expressing cancer cells and E-selectin, a calcium-dependent lectin overexpressed by microvascular endothelial cells during inflammation, is pivotal for cancer spread [19,20,21] and its inhibition has significantly improved survival in animal cancer models [22]. Accordingly, GC patients with tumors showing pronounced SLeA expression have been found at greater risk of developing distant metastasis and, consequently, face worst prognosis [23]. This antigen has also been observed in GC metastases [24], suggesting potential for designing more effective therapeutics. Even though the SLeA is not expressed by healthy blood cells, low amounts of this glycoepitope may be found in the salivary glands, colon, pancreas, bile ducts and other healthy tissues [16,25]. It has also been observed in several non-malignant conditions of the gastrointestinal tract limiting its biomarker potential [26,27,28]. Such observations challenge the cancer-specificity of the glycoantigen, urging the identification of protein glycoforms for precise targeting. As such, this work comprehensively interrogates the glycome and glycoproteome of a wide panel of GC cell models as a starting point for the identification of unique SLeA molecular signatures. It mainly focusses on understanding alterations in O-glycosylation, that is significantly altered in GC with major implications in disease outcome [14,29]. Emphasis was set on SLeA protein glycoforms with affinity for E-selectin and limited expression in healthy tissues. We anticipate that this may constitute an important roadmap towards improved disease management at different levels.

## 2. Material and Methods

### 2.1. Cell Lines and Culture Conditions

Human GC cell lines OCUM-1 and MKN-74 were purchased from the Japanese Collection of Research Bioresources (JCRB) Cell Bank, whereas NCI-N87 (N87), AGS, KATO-III and MKN-45 cells were acquired from ATCC. Cells were maintained with 10% heat-inactivated FBS (Thermo Fisher Scientific, Waltham, MA, USA) and 1% penicillin-streptomycin (10,000 Units/mL penicillin; 10,000 mg/mL streptomycin; Thermo Fisher Scientific, Waltham, MA, USA) supplemented RPMI 1640+GlutaMAXTM-I medium (Thermo Fisher Scientific, Waltham, MA, USA). OCUM-1 cells were grown with low-glucose DMEM (GE Healthcare Life Sciences, Chicago, IL, USA). Cell lines were cultured at 37 °C in a 5% CO_2_ humidified atmosphere.

### 2.2. Population and Ethics Statement

This study was performed retrospectively in a series of 202 formalin-fixed paraffin-embedded (FFPE) gastric tumor tissues obtained from archived paraffin blocks at the Portuguese Institute of Oncology—Porto (IPO-Porto), Portugal. Gastric tumors were surgically removed from 23 men and 24 women, ranging from 54 to 88 years of age (median 66 ± 12 years), admitted and treated at the IPO-Porto between 2004 and 2016. The clinicopathological information used for accessing SLeA and NCL expressions is summarized latter in this study in the results. Overall survival (OS) was defined as the period between the date of surgery and date of patient death by cancer. Clinicopathological information was obtained from patient’s clinical records, upon IPO ethics committee approval (reference 87/17 approved on 23 March 2017). Investigations were carried out following the rules of the Declaration of Helsinki of 1975, revised in 2013. A broad library of healthy tissues (appendix, skin, lung, pancreas, colon, small intestine, gallbladder, liver, testis, stomach, thyroid and tonsil) were also screened for these two markers. All procedures concerning the inclusion of patients were approved by the institutional Ethics Committee (Comissão de ética para a saúde—CES-IPOFG-EPE 87/2017) after patient’s informed consent.

### 2.3. Antibodies and Lectins

The antibodies and lectins used in this study and their applications are summarized in Table 1.

### 2.4. Flow Cytometry

Cells were detached using Versene solution (Thermo Fisher Scientific, Waltham, MA, USA), fixed with 4% paraformaldehyde (PFA; Sigma-Aldrich, St. Louis, MO, USA) and stained with mouse anti-SLeA monoclonal antibody [CA19.9-9-203] (ab116024, Abcam, Cambridge, UK) using a 1:100 dilution in PBS 2% FBS for 1 h at room temperature. Goat anti-mouse IgG (H + L) cross-adsorbed secondary antibody Alexa Fluor 488 was used for SLeA detection at a 1:300 dilution in PBS 2% FBS for 15 min at room temperature. Mouse IgG1 [MOPC-21] isotype control (ab18443, Abcam, Cambridge, UK) was included as a negative control. In parallel, 106 cells were digested with either 10 mU/mL Neuraminidase from Clostridium perfringens (Sigma-Aldrich, St. Louis, MO, USA) or 25 mU/mL PNGase F from Elizabethkingia meningoseptica (Sigma-Aldrich, St. Louis, MO, USA) at 37 °C for 1 h under mild agitation, prior to SLeA staining. PNGase F digestion assay was used to access the overall SLeA expression in glycoprotein N-glycans, whereas Neuraminidase treated cells were included as negative controls. In addition, cancer cells were screened for NCL expression using the Alexa Fluor 647-labelled rabbit anti-human NCL monoclonal antibody [EPR7952] (ab202709, Abcam, Cambridge, UK). NCL was evaluated using membrane intact live cells in the presence of propidium iodide (PI, Thermo Fisher Scientific, Waltham, MA, USA) to reassure cell surface antigen detection. Control experiments included cells permeabilized with 0.1% Triton X-100 (Sigma-Aldrich, St. Louis, MO, USA) prior to FACS analysis to determine the total NCL content. Data analysis was performed through CXP Software and results represent the standard deviation of three independent experiments.

### 2.5. O-Glycomics

Gastric cancer cell models O-glycome was characterized using Cellular O-glycome Reporter/Amplification method, as previously described [30]. A novel methodology that explores a benzyl-GalNAc residue (the first sugar in O-glycans biosynthesis) as a scaffold for the cell’s glycosylation machinery, thus enabling the identification of more extended glycans, was used in this study. After processing in the secretory pathways, secreted benzylated O-glycans could be easily recovered from the cell culture media, permethylated and analyzed by nanoLC-ESI-MS/MS. According to previous reports, this method is more sensitive than classical approaches for O-glycome characterization, which require an initial de-glycosylation step by β-elimination. Moreover, by avoiding this chemical manipulation, it reduces the probability of biased results associated with the loss of more labile sugars such as fucose and sialic acid, present in sialylated Lewis antigens. Briefly, benzyl 2-acetamido-2-deoxy-α-D-galactopyranoside (Sigma-Aldrich, St. Louis, MO, USA) was peracetylated and administered to semi-confluent cells to a final concentration of 150 μM. Following 24 h incubation, glycans were isolated from the conditioned media by filtration using 10 kDa centrifugal filter (Amicon Ultra-4; Merck KGaA, Darmstadt, Germany), followed by solid-phase extraction in Sep-Pak 3 cc C18 cartridges (Waters, Milford, MA, USA). Finally, Bn-O-glycans were permethylated and analyzed by nanoLC-ESI-MS/MS using a nanoLC system (Dionex, 3000 Ultimate nano-LC; Thermo Fisher Scientific, Waltham, MA, USA) coupled online to an LTQ-Orbitrap XL mass spectrometer (Thermo Fisher Scientific, Waltham, MA, USA) equipped with a nano-electrospray ion source (EASY-Spray source; Thermo Fisher Scientific, Waltham, MA, USA). Eluent A was aqueous formic acid (0.2%) and eluent B was formic acid (0.2%) in acetonitrile. Samples (20 μL) were injected directly into a trapping column (C18 PepMap 100, 5 μm particle size) and washed with an isocratic flux of 90% eluent A and 10% eluent B at a flow rate of 30 μL/min. After 3 min, the flux was redirected to the analytical column (EASY-Spray C18 PepMap, 100 Å, 150 mm × 75μm ID and 3 μm particle size) at a flow rate of 0.3 μL/min. Column temperature was set at 35 °C. Permethylated glycan separation occurred using a multistep linear gradient to obtain 10% eluent B at 10 min, 38% eluent B at 20 min, 50% eluent B at 55 min and 90% eluent B at 65 min. The column was maintained at 90% eluent B for 10 min before re-equilibration at 10% eluent B. The mass spectrometer was operated in the positive ion mode, with a spray voltage of 1.9 kV and a transfer capillary temperature of 250 °C. Tube lens voltage was set to 120 V and the capillary voltage to 9 V. MS survey scans were acquired at an Orbitrap resolution of 60,000 for an m/z range from 400 to 2,000. Tandem MS (MS/MS) data was acquired in the linear ion trap using a data dependent method with dynamic exclusion. The top 6 most intense ions were selected for collision induced dissociation (CID). CID settings were 35% normalized collision energy, 2 Da isolation window 30 ms activation time and an activation Q of 0.250. A window of 90 s was used for dynamic exclusion. Automatic Gain Control (AGC) was enabled and target values were 1.00e + 6 for the Orbitrap and 1.00e + 4 for LTQ MS analysis. Data were recorded with Xcalibur software version 2.1 (Thermo Fisher Scientific, Waltham, MA, USA). Glycan structures were assigned based on characteristic product ion spectra and previous knowledge about O-glycosylation pathways [30]. Glycans were expressed in terms of relative abundance in comparison to the sum of all individual contributions to the glycome.

### 2.6. Subcellular Fractionation And Protein Extraction From Gastric Tissues

Plasma membrane proteins were extracted from whole cells (20 × 106) by scrapping with fractionation buffer (20 mM HEPES buffer (pH = 7.4), 10 mM KCl, 2 mM MgCl2, 1 mM EDTA and 1 mM EGTA) on ice. Cell suspension was then passed through a 27G needle, left on ice for 20 min. and then centrifuged at 720 g for 5 min at 4 °C to remove the nuclei. The supernatants were transferred to a new tube and recentrifuged at 10,000× *g* for 5 min at 4 °C to remove mitochondria. Samples were then transferred to polycarbonate centrifuge bottles with cap assemblies and centrifuged for 1 h at 100,000 g at 4 °C. The pellets were recovered, resuspended in the fractionation buffer and passed through a 25G needle before a new centrifugation for 45 min at 100,000× *g* at 4 °C. The final pellets corresponding to membrane proteins were resuspended in an appropriate volume of TBS with 0.3% SDS. The nuclei pellets, obtained in the first centrifugation, were then passed through a 25G needle and centrifuged at 4,000× *g* for 10 min at 4 °C. Pellets were resuspended in TBS with 0.3% SDS and sonicated briefly to shear genomic DNA and homogenize the lysate. Cytoplasmic proteins obtained in the supernatant of first ultracentrifugation cycle were passed through an Amicon Ultra-4 10K Centrifugal Filter device, centrifugated at 7500× *g* for 20 min and washed extensively with fractionation buffer. The retentate was collected. The protein content in each fraction was estimated using a DC protein assay kit (Bio-Rad, Hercules, CA, USA). The purity of the fractions was estimated by Western blot using β2 microglobulin (B2M) and SLeA as biomarkers of membrane proteins, Nucleoprotein TPR (TPR) as nuclear marker, and β-actin as a cytoplasmatic/cytoskeleton marker. Proteins were also extracted from formalin fixed paraffin embedded gastric carcinoma tissues using Qproteome FFPE tissue kit (Qiagen, Hilden, Germany) according to the manufacturer’s instructions. Then the protein buffer was exchanged to RIPA buffer and the protein amount were quantified and subsequently used to access the presence of NCL-SLeA proteoforms.

### 2.7. O-Linked Glycoproteomics

The SLeA expressing glycoproteins were isolated from plasma membrane enriched extracts (200 μg) by immunoprecipitation (IP) using the anti-SLeA monoclonal antibody [CA19.9-9-203] (ab116024, Abcam, Cambridge, UK) immobilized at the surface of magnetic beads with the Pierce™ Direct IP Kit (Thermo Fisher Scientific, Waltham, MA, USA), according to the manufacturer’s instructions. In parallel, a similar strategy was used to pull-down glycoproteins with affinity for E-selectin [31]. A recombinant mouse E-selectin/ human Fc chimera (E-selectin-Ig chimera-E-Ig), a validated tool to identify E-selectin ligands in human cells [32,33], was used towards this end. The E-Ig chimera was immobilized at the surface of magnetic beads, as previously described, and incubated with the membrane protein extracts containing 2 mM CaCl2. Negative controls involving IPs with IgG1 isotype control and pulldowns with E-selectin in the absence of Ca2+ were also conducted. The glycoproteins isolated in these assays were then resolved by SDS-PAGE using 4–20% precast polyacrylamide gels (Bio-Rad, Hercules, CA, USA) and blotted for SLeA and SLeX. The bands were also excised from the gels, reduced, alkylated and digested with trypsin and identified by mass spectrometry. Tryptic digestion and nanoLC-nES-MS/MS analysis were carried out according to the conditions previously described [34]. Data was analyzed automatically using the SequestHT search engine with the Percolator algorithm for validation of protein identifications (Proteome Discoverer 1.4, Thermo Fisher Scientific, Waltham, MA, USA). Data was searched against the human proteome obtained from the SwissProt database, selecting trypsin as the enzyme and allowing up to 2 missed cleavage sites and a precursor ion mass tolerance of 10 ppm and 0.6 Da for product ions. Carbamidomethylcysteine was selected as a fixed modification, while oxidation of methionine (+15.9) was defined as variable modification. Based on glycomics studies, the database search also included as variable modifications the glycosylation of serine and threonine with HexNAc (+203.079; Tn antigen), HexNAc-Hex (+365.132; T antigen), HexNAc-Hex-Fuc (+511.190, fucosyl-T antigen); HexNAc-Hex-NeuAc (+656.228, sialyl-T antigen); HexNac(NeuAc)-Hex-NeuAc (+947.323, disialyl-T antigen), HexNAc(Hex)-HexNAc(Fuc)-Hex(NeuAc) (+1167.418, core 2 substituted with a terminal SLeA epitope). The results corresponded to proteins identified with high confidence and showing glycopeptides with SLeA as post-translational modification (PTM) in at least one cell line. MS raw files and corresponding curated results were deposited in the PRIDE—Proteomics Identification Database—EMBL-EBI (https://www.ebi.ac.uk › pride › archive) under project accession number PXD016968.

### 2.8. Bioinformatics for Biomarker Discovery

For each cell line, the subcellular location, molecular and biological functions associated with the subsets of SLeA-expressing glycoproteins with affinity for E-selectin were characterized using the Search Tool for the Retrieval of Interacting Genes/Proteins (STRING) version 10.5 (http://string-db.org/) [35]. The glycoproteins common to all the cell lines were also categorized based on transcriptomics data deposited in OncomineTM (https://www.oncomine.org/) [36] for distinct GC subtypes (intestinal, mixed, diffuse) compared to the healthy gastric mucosa. A *p* ≤ 0.05 as well as a 2-fold variation of expression were considered. This data was integrated with information on protein expression in the normal gastric mucosa from The Human Protein Atlas [37,38]. Protein expression data deposited in the Human Protein Atlas was further used to develop a scoring algorithm (target score; described here for the first time) to rank the identified glycoproteins in relation to their potential for targeted therapeutics with minimal off-target effects. To achieve this goal, the target score majorated the overexpression and localization of the glycoproteins at the cell membrane as well as associations with poor prognosis in GC, while penalizing their presence in the same subcellular compartment as in healthy tissues. Accordingly, the score system results from the sum of the following nine variables: (i) location in healthy cells (membrane: 0 points; other subcellular locations: 1 point); (ii) location in cancer cells (membrane: 1 point; cytoplasm and/or other subcellular locations: 0 points); (iii) expression in the healthy gastric epithelium at the plasma membrane (negative: 3 points; low: 2 points; moderate: 1 point; high: 0 points); (iv) expression in GC (negative: 0 points; low: 1 point; moderate: 2 points; high: 3 points); (v) prognosis value in GC (high expression is not prognostic and/or associates with favorable prognosis: 0 points; high expression associates with poor prognosis: 1 point); (vi) expression in the brain (exclusively in the brain: 1 point; brain and other tissues: 0 points); (vii) expression in lymphoid tissues at the plasma membrane (not expressed: 1 point; expressed: 0 points); (viii) expression in gametes (not expressed: 1 point; expressed: 0 points); and ix) index of expression in healthy tissues at the plasma membrane-varies from 3 points (no expression) to 0 (high expression). Interpretation of biological functions associated with glycoproteins common to all cell lines and showing affinity to E-selectin was done using ClueGO version 2.2.5 [39] and CluePedia plugins version 1.2.5 [40] for cytoscape version 3.3.0 [41]. N- and O-glycosites prediction was performed using NetNGlyc 1.0 (http://www.cbs.dtu.dk/services/NetNGlyc/) and NetOGlyc 4.0 [42], respectively.

### 2.9. Western Blot

Proteins from different sources were resolved by SDS-PAGE using 4–20% precast polyacrylamide gels (Bio-Rad, Hercules, CA, USA) and then transferred to a nitrocellulose membrane (GE Healthcare Life Sciences, Chicago, IL, USA). This included gastric cancer cell lines, total protein as well as plasma membrane, nuclear and cytoplasm proteins-enriched extracts. It also included NCL and SLeA-expressing glycoproteins isolated from membrane extracts as well as from human gastric carcinomas by immunoprecipitation, using an anti-SLeA monoclonal antibody (ab116024, Abcam, Cambridge, UK), isotype controls, anti-NCL antibody [EPR7952] (ab129200, Abcam, Cambridge, UK) or E-selectin pulldowns in the presence and absence of Ca2+ (negative control). The membranes were then blocked with 1% Carbo-Free Blocking Solution (Vector Laboratories, Burlingame, CA, USA) and probed with either the anti-SLeA (1:1,000) or the anti-NCL (1:1,000) antibodies, previously described, for 1 h at room temperature. The peroxidase affiniPure goat anti-mouse IgG (H+L) polyclonal antibody (1:90,000, 115-035-003, Jackson ImmunoResearch, West Grove, PA, USA) was used as a secondary antibody for SLeA detection. The goat anti-rabbit IgG (H+L) HPR conjugate antibody (1:60,000; G-21234; Thermo Fisher Scientific, Waltham, MA, USA) was used as a secondary antibody for NCL detection. Additional enzymatic controls were performed prior to blotting for SLeA. Briefly, membranes were incubated with 0.2 U Neuraminidase from Clostridium perfringens (Sigma-Aldrich, St. Louis, MO, USA) or 0.16 U/mL PNGase F from Elizabethkingia meningoseptica (Sigma-Aldrich, St. Louis, MO, USA) overnight at 37 °C to disclose the specificity of SLeA expression. The Amersham ECL Prime Western Blotting Detection Reagent (GE Healthcare Life Sciences, Chicago, IL, USA) was used as developing reagent.

### 2.10. Immunofluorescence Microscopy In Situ Proximity Ligation Assays (PLA)

Cells were cultured in coverslips, probed with the antibodies of interest and fixed with 4% PFA. After blocking with 5% bovine serum albumin (BSA), cells were stained with the anti-SLeA and anti-NCL primary antibodies as above described for flow cytometry. Enzymatic controls with neuraminidase from Clostridium perfringens or PNGase F from Elizabethkingia meningoseptica and permeabilization with 0.1% Triton X-100 (Sigma-Aldrich, St. Louis, MO, USA) were included. The in situ proximity ligation assay (PLA) of SLeA and NCL in cell lines was performed to detect both antigens in simultaneous proximity. The slides were incubated overnight with the primary antibodies at 4 °C. Ligation and amplification of the PLA signal were performed according to the Duolink PLA Technology kit (Sigma-Aldrich, St. Louis, MO, USA). Finally, slides were incubated with DAPI and mounted using Vectashield mounting medium (Vector Laboratories, Burlingame, CA, USA). PLA using isotype controls for one or both the antibodies of interest were used as negative controls. In addition, cells were screened regarding affinity for E-selectin using the E-selectin/human Fc chimera (E-selectin-Ig chimera-E-Ig) for 1 h at room temperature. Goat anti-mouse Alexa 549 and polyclonal rabbit anti-human IgG-FITC were used for 30 minutes at room temperature. All images were acquired on a Leica DMI6000 FFW microscope (Leica Microsystems, Wetzlar, Germany) using the Las X software (Leica Microsystems, Wetzlar, Germany) version. 

### 2.11. Immunohistochemistry

Formalin-fixed paraffin embedded (FFPE) GC tissue sections showing distinct clinicopathological features (intestinal and diffuse type GC) and healthy tissues (appendix, skin, lung, pancreas, colon, small intestine, gallbladder, liver, testis, stomach, thyroid and tonsil) were screened by immunohistochemistry for both SLeA and NCL, as previously described by us [34]. Briefly, FFPE tissue sections were deparaffinized with xylene, rehydrated with a graded series of alcohol washes and subjected to heat-induced antigen retrieval using citrate buffer pH = 6.0 (Vector Laboratories, Burlingame, CA, USA) for 15 min in the microwave, after pre-heating of the solution at maximum power rating for 5 min. Sections were incubated with 0.3% hydrogen peroxide (Merck KGaA, Darmstadt, Germany) for 25 min, blocked with UV Block® (Thermo Fisher Scientific, Waltham, MA, USA) and incubated overnight at 4 °C in a wet chamber with either anti-SLeA (1:100 5% BSA-PBS) or anti-NCL (1:250 5% BSA-PBS) antibodies. After washing with PBS-Tween, biotinylated secondary antibody was added to tissue sections, before incubation with streptavidin (Thermo Fisher Scientifc, Waltham, MA, USA). Antibodies’ binding was detected by incubation with 3,3′-diaminobenzidine (ImmPACT™ DAB, Vector Laboratories, Burlingame, CA, USA) for 4 min. Nuclei were counterstained with hematoxylin for 1 min. Positive and negative control sections were tested in parallel. Negative control sections were performed using 5% BSA-PBS devoid of primary antibody. Positive controls consisted of known positive tumor tissues for the antigen in study. Tumors were classified as positive when immunoreactivity was observed by microscopic presence of brown chromogenic product in tumor cells. Antibody staining was assessed double-blindly by two independent observers and validated by an experienced pathologist. Whenever there was a disagreement, the slides were reviewed, and consensus was reached. To evaluate sialic acid-dependent binding of the antibodies, tissues were treated with sialidase prior to analysis.

### 2.12. Statistical Analysis

All experiments were independently replicated at least as three times. Statistical data analysis was performed with IBM Statistical Package for Social Sciences—SPSS for Windows (version 20.0, IBM, Armonk, NY, USA). One-way ANOVA and Mann–Whitney U test were used. Chi-square analysis was used to compare categorical variables. Kaplan-Meier curves were used to analyze the influence of the biomarkers expression in the context of time-to-event (death). Comparison of estimates was done using log-rank test. A 95% significance threshold for the null hypothesis was considered *p* < 0.05 (* *p* < 0.05, ***p* < 0.01, ****p* < 0.001). A cluster heatmap analysis was conducted to compare the cells lines in relation to their glycomics patterns. The analysis was carried out in MetaboAnalyst 4.0 [43] using log-transformed and auto-scaled data. Cluster analyses followed the criteria (i) cluster rows and columns; (ii) similarity metrics: Euclidean distance; and (iii) clustering method: Ward.

## 3. Results

The expression of SLeA at the cell surface is decisive for E-selectin mediated hematogenous dissemination of GC cells and the formation of local and distant metastasis [14,29]. However, the proteoforms carrying this modification in GC cells remain to be elucidated, which is a critical aspect for precise targeting of subpopulations showing high metastatic potential. As such, this work started by screening for SLeA a panel of cell models reflecting distinct aspects of the disease (Appendix A). The overall goal was to elect models for downstream targeted glycoproteomics studies, foreseeing the identification of SLeA-expressing glycoproteins showing high affinity for E-selectin. Emphasis was set on disclosing alterations in the O-glycoproteome, which is well known to experience significant remodeling during GC progression and dissemination [14,44].

### 3.1. Glycomics and Affinity for E-selectin

Six cancer cell models were screened for SLeA expression by flow cytometry (Figure 1A,B). This showed high heterogeneity in SLeA expression, most likely reflecting different glycosyltransferases repertoires [45]. Namely, AGS and MNK-74 cells did not express SLeA, whereas high levels of this glycan were observed in N87 and OCUM-1 (60–80% of positive cells) and, to a much lesser extent, in MKN-45 and KATO-III cells (>20%). Subsequently, cells were N-deglycosylated with PNGase F to determine the origin of the SLeA antigen. This had little impact in MKN-45, KATO-III and OCUM-1, strongly suggesting that SLeA may be mostly derive from O-glycans in glycoproteins and, potentially, glycolipids (Figure 1A–C). On the other hand, exposure to PNGase F significantly decreased SLeA levels in N87 cells (4-fold), demonstrating that in this cell line most of the antigen comes from N-glycans. Nevertheless, 20% of cells retained SLeA expression after enzymatic treatment, also suggesting significant O-glycosylation levels. Notably, the signals in OCUM-1 and N87 cells were highly responsive to neuraminidase digestion, demonstrating the specificity of the anti-SLeA antibody for sialylated glycans (Figure 1A–C). In parallel, fluorescence microscopy studies demonstrated that recognition of OCUM-1 and N87 cancer cells by E-selectin was dependent on the presence of SLeA and increased with the levels of the antigen (Figure 1D). Control experiments conducted in the absence of Ca2+ (required for E-selectin binding) as well as with cells previously digested with neuraminidase did not show relevant affinity of the lectin for cancer cells, reinforcing previous observations.

We then comprehensively interrogated the cells’ glycome by mass spectrometry with the objective of validating these results. This was also used to gain more insights on the structure of the glycans exhibiting sialylated Lewis antigens as terminal epitopes, which was of key importance for protein annotation upon glycoproteomics characterization. Based on flow cytometry analysis and previous reports, emphasis was set on the O-glycome [44]. As highlighted by Figure 2A, 23 distinct glycans were identified. Overall, the glycome of all cell lines was composed by core 1 and core 2 fucosylated and/or sialylated O-glycans of variable lengths. Nevertheless, shorter glycans predominated over more extended structures beyond core 2 (Figure 2A,B) and all cell lines, irrespectively of their origin, overexpressed the T antigen (core 1; m/z 572.3), fucosylated-T antigen (Fuc-T; m/z 746.4), Sialyl-T antigen (ST; m/z 933.5) and short core 2 structure (m/z 991.5; 1021.5; 1195.6) antigens. However, key differences could be identified when zooming in on the glycome of each cell line. According to the heat map on Figure 2B, OCUM-1, AGS and KATO-III cells were enriched for neutral glycans, including core 1, Fuc-T and short core 2 structures (m/z 1021.5), exhibiting considerable degrees of mono- (m/z 1195.6), di- (m/z 1369.7; 1818.9) and tri-fucosylation (m/z 1543.8). On the other hand, MKN-45, MKN-74 and N87 cells presented significant oversialylation (e.g., m/z 933.5, 1178.7, 1294.7). MKN-45 exhibited more extended fucosylated and/or sialylated glycans, while MKN-74 were significantly enriched for the sialyl-T antigen and N87 cells presented shorter mono- and di-sialylated core 1 and core 2 glycans and the sialyl-Tn (STn antigen; m/z 729.4), a glycan known to play a critical role in GC invasion and metastasis [46,47]. Despite marked differences, MKN-45, KATO-III, OCUM-1 and N87 also exhibited a low abundance of ions (<5% relative abundance; m/z 1556.8, 1801.9 and 1917.9), potentially corresponding to glycans with terminal SLe epitopes, in accordance with flow cytometry analysis. Moreover, the chromatographic profiles for these samples suggested the presence of SLeA and SLeX antigens (Figure 2C), which could not be observed in SLe negative cell lines (AGS and MKN-74). Product ion spectra showing cross-ring fragmentations characteristic of both isomers (Figure 2D) allowed the annotation of chromatographic signals corresponding to SLe antigens (Figure 2C) and a relative estimation for SLeA in relation to SLeX in each cell line (Figure 2E). In particular, the SLeA product ion spectrum (left spectrum in Figure 2D) presented diagnostic signals resulting from different cross-ring cleavages (0,3X, 1,3X or 3,5X; carbohydrate fragmentations according to Domon and Costello [48]) at GlcNAc residues (*m/z* = 554.5, *m/z* = 707.7, *m/z* = 815.6 and *m/z* = 1061.9). The fragment ion at *m/z* = 653.9 (Z3,5XGlcNAc), not observed for SLeA, defined SLeX (right spectrum in Figure 2D). Based on these observations, N87 cells expressed similar amounts of both isomers, whereas OCUM-1, KATO-III and MNK-45 mostly overexpressed SLeA and minor amounts of SLeX (<10%). The presence of SLeX in these cell lines was further confirmed by flow cytometry). In summary, glycomics characterization provided important structural information for guiding future studies on the functional role of O-glycans in GC. Moreover, it demonstrated that SLeA is not a major determinant of the O-glycome, despite its tremendous impact on metastasis development. Nevertheless, SLeA was found as the dominant sialylated Lewis O-glycoform in GC cells; even though the presence of its isomer SLeX could also be detected. Taken together with flow cytometry analysis (Figure 1C), our findings support that SLeA may be expressed by subpopulations of cells, reinforcing the importance of engaging in targeted glycoproteomics studies towards their precise identification. In this context, glycomics analysis was also crucial for the identification of glycan modifications to be used in targeted glycoproteomics, enabling higher protein coverage and validation of relevant glycosylated patterns. 

### 3.2. Targeted Glycoproteomics and Glycobiomarker Discovery

We then devoted our attention to characterizing the SLeA glycoproteome of different cancer cell models with the goal of identifying common molecular signatures for precise cancer targeting. OCUM-1, N87 and KATO-III cell lines, exhibiting relevant SLeA levels, were elected for this study. Glycoproteins were isolated from extracts enriched for plasma membrane proteins by immunoprecipitation with an anti-SLeA monoclonal antibody (scheme in Figure 3A). These were subsequently resolved by SDS-PAGE and identified using a conventional bottom-up proteomics strategy (excision of gels spots, reduction, alkylation and digestion with trypsin) culminating in nanoLC-nES-MS/MS analysis. Protein annotations were performed including the possibility of variable protein modifications with the Tn antigen (the simplest form of O-glycosylation) as well as the T, ST, Fuc-T and extended core 2 substituted with SLe antigens, in accordance with glycomics analysis. The final list of identifications included proteins annotated with high confidence, irrespectively of the SLeA status. Prior to analysis, we evaluated the SLeA antigen by Western blot in IP samples from both SLeA+ (N87) and SLeA- (AGS) cell lines (Figure 3B). Samples obtained with the corresponding isotype control were also screened. As expected, SLeA was not detected in the isotype controls nor in SLeA- cell lines (Figure 3B). In parallel we also evaluated the potential recovery of SLeX together with SLeA-expressing glycoproteins by this method. All blots were negative for SLeX (Figure 3B), reinforcing the specificity of the isolation protocol for SLeA expressing glycoproteins. In parallel, glycoproteins with affinity for E-selectin were pulled-down with an E-selectin-IgG chimera and identified, to disclose potential mediators of metastases development. Prior to MS/MS analysis, E-selectin pulldowns were screened for SLeA expression by Western blot and compared to samples obtained in the absence of Ca2+ ions (negative controls). Again, SLeA could only be observed in SLeA positive cell lines and only when pulldowns were performed in the presence of Ca2+ (Figure 3B). Moreover, as expected, SLeX could be detected in SLeA positive extracts of N87 cell lines, since the lectin recognizes both glycoepitopes (Figure 3B). Collectively, preliminary experiments support the specificity of the SLeA IP for glycoproteins presenting this glycan and the affinity of E-selectin pulldowns for glycoproteins expressing both SLeA and SLeX.

MS/MS analysis after immunoprecipitation assays for the SLeA antigen identified 889, 344 and 665 proteins in N87, OCUM-1 and KATO-III cell lines, respectively (Figure 3C; raw data and assignment tables available at PRIDE accession number PXD016968). Interestingly, few glycoproteins (approximately 10% of the total identified glycoproteins for each cell line) were common to all cell lines, denoting highly cell-specific SLeA-glycoproteomes (Figure 3C). However, the number of glycoproteins showing affinity for E-selectin was lower in comparison to SLeA-immunoprecipitation assays (630 proteins for N87, 309 for OCUM-1 and 500 for KATO-III). Nevertheless, approximately 60% of the identified glycoproteins were simultaneously identified by the two strategies for each cell line (532 for N87, 186 for OCUM-1, 367 for KATO-III) (Figure 3C). Moreover, MS/MS analysis confirmed the presence of O-glycopeptides with SLeA epitopes in more than 20% of the identified glycoproteins in OCUM-1 and N87 cell lines but just 1% in KATO-III, in agreement with glycomics studies showing lower SLeA expression in this cell line. Moreover, more than 90% of the glycoproteins identified in these studies presented glycosylation patterns characteristic of cell surface glycoproteins (PRIDE accession number PXD016968), in accordance with their cell surface origin. However, we have also noted that a significant percentage of the glycoproteins isolated by SLeA IP did not show affinity to E-selectin (approximately 40% for all cell lines). Even though the existence of non-specific immunoprecipitations is frequent and may decisively contribute to these observations, approximately 50% of the glycoproteins in the SLeA-IP group, which could not be found using E-selectin, exhibited SLeA-glycopeptides. Such observations strongly suggest that the expression of sialylated Lewis antigens is not the unique pre-requisite for E-selectin binding. It is possible that the nature of the glycoproteins presenting these modifications, as well as the density and structural organization of glycans chains, may also influence this process, which warrants a more profound evaluation in future studies. On the other hand, there is a significant portion of E-selectin identifications that do not match in the SLeA-IP group. The expression of SLeX by these cell lines (Figure 2C–E), which is also a key ligand for E-selectin [15], may account for these findings. 

We then used gene ontology to comprehensively classify SLeA-expressing glycoproteins with affinity for E-selectin, according to their class (Panther), main biological functions and associated subcellular locations (ClueGo and CluePedia). Accordingly, Appendix A shows a wide number of different protein classes for each cell line (20 for N87, 11 for OCUM-1, 14 for KATO-III), denoting significant differences between the distinct cell models. Nevertheless, the main represented protein classes included signaling and nucleic acid binding for N87 and OCUM-1/Kato-III cell lines, respectively. Strikingly, many of these glycoproteins (20–60%) were primarily associated with protein translation and biosynthesis in ribosomes, spliceosome complexes and even at the nuclear euchromatin. Proteins typically found in the cytoplasm and the cytoskeleton were also detected (10–20%). On the other hand, only 10–15% of them could be associated with the plasma membrane, cell junctions and the extracellular space, known to typically undergo O-glycosylation across the secretory pathways. While collectively these findings could question the quality of glycoproteins isolation methods, the presence of glycopeptides with terminal SLeA glycans in all identified glycoprotein challenges this concept, suggesting that a significant number of proteins may experience subcellular miss-localization and appear at the cell membrane yielding abnormal glycosylation patterns. Although this is a defying concept, proteins miss-localization has been long described and frequently observed for more aggressive cancer cells, generally associated with more aggressive cancer subpopulations [49,50,51,52]. Such events may most likely explain current observations but warrant confirmation in future studies.

In summary, we have demonstrated the existence of a wide array of glycoproteins showing high affinity for E-selectin, including many glycoproteins with confirmed O-SLeA glycoforms (193 glycoproteins). These 193 glycoproteins were distributed among the three cell lines according to the Venn Diagram in Figure 3C. The N87 model presented a more unique glycopattern in comparison to the other cell lines (95 unique glycoproteins in N87 vs. 4 for OCUM-1 and 1 for KATO-III). A high degree of homology was also observed between N87 and KATO-III cell lines at this level (55 common glycoproteins). More importantly, only 22 protein species were common to all cell lines (Figure 3C), suggesting a relatively low degree of similarity between the three cell models. Nevertheless, the identified glycoproteins constituted an important starting point for identification of common molecular grounds potentially associated with metastases development. 

Envisaging molecular signatures of potential clinical interest for a wide number of tumors, the focus was placed on the restricted group of 22 glycoproteins found in all cell lines. According to transcriptomics data deposited on the Oncomine repository, only 5 glycoproteins were significantly overexpressed in GC (KRT5, HIST1H1D, PSMD2, ANXA2 and SPTAN1), distributed between the different types of gastric tumors, according to the Lauren classification (intestinal, diffuse and mixed type adenocarcinomas), as highlighted in Figure 4A. Gene expression was then matched with the levels of the proteins in gastric tumors, as defined by the Human Protein Atlas (Figure 4B). Notably, HIST1H1D, PSMD2, ANXA2 and SPTAN1 overexpression was associated with elevated protein content in cancer tissues, whereas KRT5 has been rarely observed at the protein level in these tissues (Figure 4B). HIST1H1D overexpression was observed in all three types of gastric lesions, whereas PSMD2 was mainly linked to intestinal and mixed type tumors, ANXA2 to intestinal and diffuse subtypes and SPTAN1 to the mixed subtype. Nevertheless, the functional implications of these glycoproteins in the context of the different types of lesions is poorly understood and should be subject to more profound investigation in future studies. Moreover, all these glycoproteins were also tremendously overexpressed in healthy tissues, limiting their potential for targeted therapeutics. 

To tackle the difficulties associated with cancer biomarkers specificity, we applied an alternative strategy using an algorithm developed in-house exploiting information from the Human Protein Atlas. This algorithm, termed target score, was designed to pinpoint cancer biomarkers showing high cancer specificity and limited off-target potential due to low/null expression in healthy tissues (Figure 4C). It balanced the expression of proteins in healthy tissues versus cancer; its subcellular localization in tumors, privileging expression at the cell surface; and associations with poor prognosis. On the other hand, it severely penalized proteins showing wide distribution in healthy organs. Surprisingly, the top-ranked protein among the 22 glycoproteins common to all cell lines was nucleolin (NCL; Figure 4C), a protein generally found in the nucleus of healthy cells, but previously reported at the cell membrane of esophageal [53], gastric [51], colorectal [54] and breast [55] cancer cells. Moreover, according to analysis in silico, NCL may be densely O-glycosylated (23 O-glycosites; NetOGlyc 4.0) and present also 3 N-glycosites (with NetNGlyc 1.0) (Appendix A), supporting its detection using mass spectrometry-based immunoassays directed for glycans. Interestingly, the second top-targeted glycoprotein X-ray repair cross-complementing protein 6 (Ku70; Figure 4C) was also a typical nuclear protein poorly studied in GC. Like NCL, the Ku70 protein has been previously found at the cell membrane of cancer cells in other models, playing a key functional role driving extracellular matrix remodeling and invasion [56]. The Ku70 protein also showed a higher density of potential O-glycosylation sites (*n* = 11) in comparison to N-glycosites (*n* = 1) upon bioinformatics analysis. Collectively, these findings reinforce the relevance of addressing O-glycosylation for biomarker discovery and support the notion of protein misslocalization in cancer cells. Contrastingly, most of the glycoproteins pinpointed as significantly overexpressed in gastric tumors according to transcriptomics and immunohistochemistry using monoclonal antibodies (highlighted by Figure 4A,B) were found in the lower part of the ranking, severely penalized by its high levels of expression in healthy tissues, suggesting low potential for precise cancer targeting. Such findings reinforce the importance of the target score system as a valuable tool for identifying unforeseen biomarkers at the cell membrane. The target score’s sensitivity for protein misslocalization pinpointed several novel targetable molecules that were not captured by transcriptomics and conventional immunoassays (Figure 4), constituting a decisive new tool for glycoproteome datamining and biomarker discovery.

Finally, we explored protein–protein networks provided by ClueGo and CluePedia for Cytoscape to gain more insights on their potential functional roles. In general, there is a significant level of interaction between the identified glycoproteins in the mediation of six main biological functions: i) differentiation; ii) cytoskeleton structure; iii) cadherin-mediated cell–cell adhesion; iv) protein targeting to the cell membrane; v) protein translation; and vi) cellular response to epidermal growth factor stimulus (Figure 4D). Such observations support a conserved role for these glycoproteins in distinct cell models. Notably, NCL constituted an important hub protein in the regulation of protein translation (Figure 4D), most probably when present in the nucleus. Surprisingly, it was also observed as a mediator of response to epidermal growth factor, which is a key aspect in GC underlying progression and dissemination [57,58]. Although the functional role of NCL in GC remains unknown, important observations were made to guide future studies on this field. 

Collectively, we comprehensively interrogated the O-glycoproteome of GC cells for common molecular signatures showing affinity for E-selectin using different bioinformatics approaches. Strikingly, NCL emerged as a top-ranked potentially targetable protein at the cell membrane and a potential ligand for E-selectin. More efforts were then devoted to validating the presence of this glycoprotein at the cell membrane, its substitution with glycans presenting SLeA and providing a clinical context for its appearance in GC.

### 3.3. Nucleolin Expression in Gastric Cancer Cells

Immunofluorescence staining showed that NCL was not present in cell nucleus and seems to be distributed along cytoplasm and cell membrane in OCUM-1, KATO-III and N87 cells (Figure 5A). Furthermore, both NCL and SLeA seem to be in close proximity, as reinforced by the positive signal in situ PLA (Figure 5A,B). Notably, the low density of red dots denoting positive PLA suggests that NCL-SLeA proteoforms may be restricted to specific cells, possible endowed with high metastatic capacity, which warrants future clarification. On the other hand, the lack of antibodies targeting specifically NCL at the cell membrane may significantly contribute to its underestimation. Nevertheless, the presence of NCL at the cell membrane was also supported by flow cytometry and immunofluorescence analysis before and after membrane permeabilization (Appendix A). In contrast, SLeA negative cell models (AGS and MKN-74) showed just residual amounts of NCL at the cell membrane and presented more elevated intracellular NCL in comparison to the other models, supporting a close association between these events. Future efforts should be focus on understanding the role of glycosylation in stabilization of this glycoprotein at the cell membrane and underlying molecular mechanism supporting subcellular compartment migration.

The nature of NCL glycoforms was then addressed by tandem mass spectrometry analysis (Figure 5C,F). The glycoproteins resulting from the SLeA and E-selectin/IgG chimera IPs were analyzed in a molecular weight-resolved manner. NCL was detected above 100 kDa, most likely corresponding to heavily glycosylated proteoforms of its canonical form at 76 kDa. However, it was also detected between 75 and 50 kDa in all cell lines and at lower molecular weights (37–15 kDa), which was particularly evident for OCUM-1 and KATO-III (Figure 5F). Unequivocal annotations were confirmed by O-glycopeptides containing the SLeA, as highlighted by the MS/MS spectra in Figure 5E showing product ion spectra combining fragments ions from the peptide and the glycan backbones (m/z 366.2, 731.3, 877.3 and 1167.42). Western blot analysis for SLeA IPs confirmed the patterns identified by MS/MS (Figure 5F), reinforcing that GC cells may express a wide array of NCL glycoproteoforms of distinct lengths. For cross-validation, NCL IPs were probed for SLeA and screened for E-selectin affinity before and after neuraminidase digestion (negative control) (Appendix A), which also confirmed mass spectrometry assignments. Equally important, these identifications matched with E-selectin IPs (Figure 5D), suggesting that E-selectin may target distinct NCL glycoproteoforms, irrespectively of their size.

To disclose the potential existence of cell membrane NCL proteoforms, as suggested by glycoproteomics, we performed subcellular fractionation of N87, OCUM-1 and KATO-III cells. Plasma membrane, nucleus and cytoplasmatic protein enriched extracts were then screened for NCL (Appendix A). The purity of each extract was assessed based on the expression of biomarkers associated with each subcellular location: SLeA and B2M (cell membrane); TPR (nucleus); β-actin (cytoplasm). Collectively, results suggested that both plasma membrane and cytoplasmatic extracts presented a considerable degree of purity. More importantly, membrane protein fractions appear not to present nuclear proteins, further supporting the existence of plasma membrane NCL. Strikingly, all subcellular fractions share two main NCL bands at 100 and just above 50 KDa, suggesting that these proteoforms may be present in all subcellular compartments. Nevertheless, a considerably higher number of other proteoforms could be identified in the plasma membrane and the nucleus, which were not seen in the cytoplasm. Moreover, there are marked differences between the plasma membrane and the nuclear fractions, despite its contamination with glycosylated proteins. Even though more effective subcellular fractionation methods would be required, these observations support the existence of specific NCL proteoforms associated with the cell membrane. The future identification of such proteoforms will be required for better understanding the functional role of this protein at the cell surface. 

### 3.4. SLeA and NCL Expression in Gastric Cancer and Healthy Tissues

We then screened a large patient cohort (202 cases; Table 2; Appendix A) and 56 lymph node metastases for SLeA expression. The SLeA antigen was detected in approximately 82% of the tumors, corresponding to more than 30% of the tumor area in 64% of the cases (SLeA overexpression), irrespective of its clinicopathological classification (Figure 6A, Table 3). It showed an intense cell membrane expression in cancer cells but could also be observed in the cytoplasm, most likely associated with glycoproteins across the secretory pathways. Notably, SLeA signals in both primary tumors and metastases were responsive to digestion with neuraminidase but showed little changes after incubation with PNGase F (Appendix A), supporting the key contribution of O-glycans suggested during glycomics studies. Equally importantly, the SLeA antigen was overexpressed by 71% of the lymph node metastases, showing significant correlation with its presence in the primary tumor (*p* = 0.006), in agreement with its metastasis-associated nature. Moreover, the SLeA antigen was exclusively found in the tumor area but not in the surrounding apparently normal mucosa, denoting some degree of cancer specificity. According to Table 3, SLeA overexpression was significantly associated with more advanced stages of the disease (*p* = 0.002) and the presence of distant metastases, reinforcing its association with more aggressive phenotypes and key role in disease dissemination. Moreover, SLeA overexpression showed a trend association with decreased OS in GC (*p* = 0.07). 

We then evaluated NCL in a subset of 47 gastric tumor sections comprehending distinct clinicopathological types (23 intestinal, 24 diffuse) as well as 7 lymph node metastases, thus reflecting a clinicopathological nature (Table 3), similar survivals as the main cohort (Appendix A), and showing SLeA expression patterns very similar to the main patient cohort. The main objective was to disclose the strong cancer associated nature of NCL-SLeA glycoforms suggested by glycomics, glycoproteomics and comprehensive bioinformatics studies. The option for a smaller patient subset was directly linked to the very time-consuming nature of NCL subcellular evaluation facing extensive nuclear expression and the lack of specific antibodies for NCL glycoforms. Nevertheless, the evaluation of these tumors showed that NCL was extensively expressed in the nucleus of most cells in all cases (Figure 6A). However, half the tumors also presented scattered cancer cells (not exceeding 5% of the lesion) with marked membrane expression accompanied by the loss of nuclear expression (Figure 6A). The NCL membrane phenotype was also observed in 57% (4/7) of lymph node metastases, suggesting a possible role in this process. However, neither SLeA nor NCL at the cell membrane showed associations with clinical variables (Table 3) in this patient subset, most likely due to the number of analyzed cases. In addition, we observed that NCL expression at the cell membrane and SLeA occurred in the same tumor areas in approximately 40% of the cases, suggesting NCL glycophenotypes like GC cell lines. To confirm this hypothesis, we immunoprecipitated SLeA expressing glycoproteins from both intestinal and diffuse type gastric tumors. The Western blots in Figure 6B support the existence of NCL-SLeA in GC and show that shorter proteoforms at approximately 37 kDa predominate in relation to higher molecular weight species. Again, no associations were found between the NCL-SLeA and clinicopathological variables, which warrants validation in a larger cohort. However, we observed a statistically significant association between NCL-SLeA and decreased OS (*p* = 0.05; Figure 6C), which was not reflected by membrane NCL alone (*p* = 0.405; Figure 6C). These observations reinforce the importance of targeting abnormal protein glycosylation towards clinically useful biomarkers. In summary, we identified a subset of gastric tumors expressing NCL-SLeA glycoforms and facing worst prognosis, supporting the importance of targeting glycosylation for more precise cancer biomarkers. We also observed that this glycophenotype is characteristic of minor subpopulations of cells with yet unknown role in the disease. Future studies must be devoted to comprehensively disclosing the clinical significance of NCL-SLeA in larger patient cohorts, foreseeing improved clinical management and potentially targeted therapeutics. Emphasis should also be placed on conducting studies in the context of currently accepted molecular disease subtypes [59]. It is also pressing to address the functional impact of these molecular features for cancer cells. 

Foreseeing precise cancer targeting, we evaluated SLeA and NCL in a wide array of relevant healthy tissues. SLeA was observed in the cytoplasm of exocrine pancreas cells, suggesting association to secreted proteins (Figure 6D). It was also mildly detected at the cell membrane of goblet cells of the gastrointestinal tract, cells composing the upper layers of the corneal epithelium, rare thyroid parafollicular cells and pulmonary secretions (Figure 6D). In turn, nuclear NCL was abundantly observed in all tissues. On the other hand, membrane protein expression was observed in corneal epithelium, not in the same cell types expressing SLeA, and epidermis (Figure 6). Additional studies on membrane NCL and SLeA positive tissues also retrieved negative These observations suggest that the conjugation of SLeA and NCL at the cell membrane is a rare event associated with cancer, holding cancer-specific targeting potential. Nevertheless, screening a wider array of healthy tissues is still required to fully characterize NCL-SLeA glycoforms expression patterns in health and disease. 

## 4. Discussion

The tremendous advances in the molecular subtyping of gastric tumors over the past five years have significantly contributed to improving its clinical management, enabling better therapeutic selection and prognosis [9,10,60]. However, the quest for molecular signatures associated with metastatic spread persists as a daunting research topic due to significant intra- and inter-tumor molecular heterogeneity. Exploring alterations in protein glycosylation occurring at the cancer cell surface holds tremendous potential towards this end [13,14]. 

Based on these observations, we focused on glycoproteins expressing the SLeA antigen, a well-known mediator of metastasis that allows GC cells to bind E-selectin on activate endothelial cells, supporting their extravasation into the blood and lymph and, ultimately, intravasation into several organs [18,61,62,63]. Accordingly, we have identified the SLeA in both N- and O-glycoproteins and specific O-glycome signatures associated with this modification. Nevertheless, despite its key functional role, SLeA was residually expressed in comparison to other types of O-glycans, denoting a highly restricted phenotype, possibly associated with more aggressive cell subpopulations, as suggested by different studies [25,64]. This structural information also enabled the identification of a wide number of glycoproteins, significantly broadening our understanding on the SLeA-glycoproteome. Glycoproteomics studies clearly showed that the SLeA-glycoproteome varies significantly with the type of cell and highlighted marked cell-type dependent glycoproteomes, reinforcing the need for personalized molecular characterization. However, only a small percentage of the identified glycoproteins showed affinity for E-selectin, suggesting that the presence of SLeA is not a sole determinant for this process. Future studies must comprehensively disclose how the density and distribution of glycosites and the structure and type of glycans chains, as well as the nature of the protein, may influence E-selectin recognition. Future studies should also focus on characterizing the SLeX-glycoproteome, equally targeted by E-selectin and widely expressed in GC [65,66]. These aspects will be critical for elucidating the molecular basis mediating the interaction between GC cells with endothelial cells. Notably, sialylated Lewis glycan mimetics are currently being used in pre-clinical and clinical settings to inhibit E-selectin mediated metastasis [22,67]. A more profound knowledge about the molecular aspects governing this interaction will be a key milestone towards improving this emerging therapeutic strategy. 

Finally, we observed that only a small subset of 22 SLeA-expressing glycoproteins with affinity for E-selectin were conserved in all cell lines. An in-house target score algorithm was developed and used for the first time to rank these glycoproteins according to their cancer-specificity. The objective was to establish a proteome interrogation tool capable of capturing cancer specific biomarker signatures not easily foreseen by other omics. Surprisingly, NCL, a typical nuclear protein found generally associated with intracellular chromatin and playing a central role in polymerase I transcription in healthy cells [68], emerged as a top-ranked cancer protein at the cytoplasm and cell membrane. Its location in cancer cells not restricted to the nucleus, as generally occurs in healthy cells, significantly contributed to this stratification. Interestingly, the presence of NCL at the cell surface was described for the first time over thirty years ago [69] and has since been observed in cancer cells of different origins [68], including in GC [70,71]. Nevertheless, these observations remain quite surprising, since the protein does not present transmembrane hydrophobic domains nor a plasma membrane targeting sequence [68]. Moreover, it has been found glycosylated by an as-yet unraveled molecular mechanisms [72,73], with glycosylation playing a fundamental role in its biological function [72,73,74]. Namely, at the plasma membrane, NCL interacts with several membrane and external ligands involved in cell proliferation, differentiation, adhesion, mitogenesis and angiogenesis [68]. In GC, membrane NCL has also been found to mediate the incorporation of tumor necrosis factor-α (TNF-α)-inducing protein, produced in large amounts by Helicobacter pylori. The NCL-TNFα-inducing protein complex was shown to promote epithelial-mesenchymal transition of cancer cells, inducing cell migration and invasive morphological changes [75]. On the other hand, in the nucleus, NCL plays a relevant role maintaining genome stability [76]. Therefore, in cancer cells, nuclear NCL overexpression is considered critical to control the balance between DNA replication and repair, limiting DNA damage accumulation due to high proliferation rates [68]. Supporting these observations, high levels of nuclear NCL was demonstrated to be an independent prognostic marker for better survival in GC, while high cytoplasmic staining was closely associated with worse prognosis [77]. Nevertheless, the clinical significance of its translocation to the cell membrane has not yet been disclosed in the context of GC. However, it is possible that NCL abnormal subcellular localization at the cell membrane may promote a more aggressive phenotype, as suggested by its functional role in GC. Notably, our bioinformatics analysis associated NCL, together with other identified glycoproteins, to cellular responses to epidermal growth factor stimulus. Supporting these observations, NCL has been found to stabilize ErbB1 and interact with Ras. The three proteins act synergistically to mediate tumor growth and favor cancer cell proliferation and survival [78,79]. Moreover, bioinformatics analysis suggest that a similar mechanism may occur in GC, warranting further investigations to understand the exact mechanism and clinical implications.

Another striking observation is that NCL at the cell membrane is targeted by E-selectin, strongly suggesting its glycoforms may play a key role in disease dissemination, which should be addressed in future studies. These glycoforms were of low molecular weight compared to the canonical form. It is likely that altered splicing mechanisms accompanying disease progression may lead to the expression of truncated forms, which warrants also careful evaluation envisaging to understand their functional role in cancer. Importantly, NCL-SLeA glycoforms showed significant associations with worst overall survival in GC which were not evident based on NCL evaluation alone. These findings support the importance of targeting specific protein glycoforms in order to improve the cancer-specificity of relevant biomarkers, as previously demonstrated by us for other models [34]. Equally important, NCL-SLeA glycoforms were detected in the primary tumors as well as the corresponding metastases, supporting a possible role in disease dissemination mediated by SLeA-E-selectin interactions [80]. Although the interaction between NCL and E-selectin via SLeA is being suggested for the first time, NCL has been recently found to bind L-selectin, another type of selectin expressed by lymphocytes that also targets sialylated antigens. In this case, NCL was described as a major ligand mediating head-and-neck squamous carcinoma cells to lymphocytes, under low shear stress conditions, required for successful adhesion of cancer cells to the lymphatic vascular compartment [81]. Collectively, these findings support that NCL carrying sialylated antigens may help promoting cancer metastasis at different levels. Moreover, it may also contribute to immune evasion by fostering the adhesion of cancer cells to platelets via P-selectin [12,82], which also binds to sialylated Lewis antigens. While the functional dimension of these observations remains to be fully clarified and should be subjected to future investigations, it becomes clear that NCL-SLeA glycoforms hold potential for addressing both intestinal and diffuse tumors and associated metastases. Envisaging this goal, we have screened a wide number of relevant healthy tissues for membrane NCL and SLeA antigen. According to our observations, sole targeting of membrane NCL or SLeA could potentiate off-target effects negatively affecting ocular, cardiac and gastrointestinal organs. Contrastingly, the bi-specific nature of NCL-SLeA glycoforms may pave the way for precise cancer targeting. 

In summary, this work sets a novel molecular rationale for targeting GC cells of higher metastatic potential translated by increased affinity for E-selectin. It also suggests that the overexpression of NCL-SLeA glycoforms at the cell surface of cancer cells may be a key mediator of this process, which warrants confirmation by studies in vitro and in vivo. The cancer-specific nature of these alterations further supports potential for precise cancer targeting with limited off-target effects in healthy organs. It is now becoming crucial to elucidate the functional impact of such observations as well as their clinical context foreseeing improved strategies for controlling disease dissemination. 

## 5. Conclusions

This work shows that different types of gastric cancer cells express NCL at the cell membrane expressing O-glycans terminated with the SLeA antigen that may be recognized by E-selectin. Moreover, NCL-SLeA glycoforms are expressed in the primary tumors and metastases, are associated with worst prognosis, and could not be found in healthy human tissues, supporting the potential for targeted therapeutics. To our knowledge, this is the first report describing SLeA in preferentially nuclear protein, setting a new paradigm for cancer biomarkers discovery. Future studies should devote to the validation of these findings foreseeing precision oncology.

## Figures and Tables

**Figure 1 cancers-12-00861-f001:**
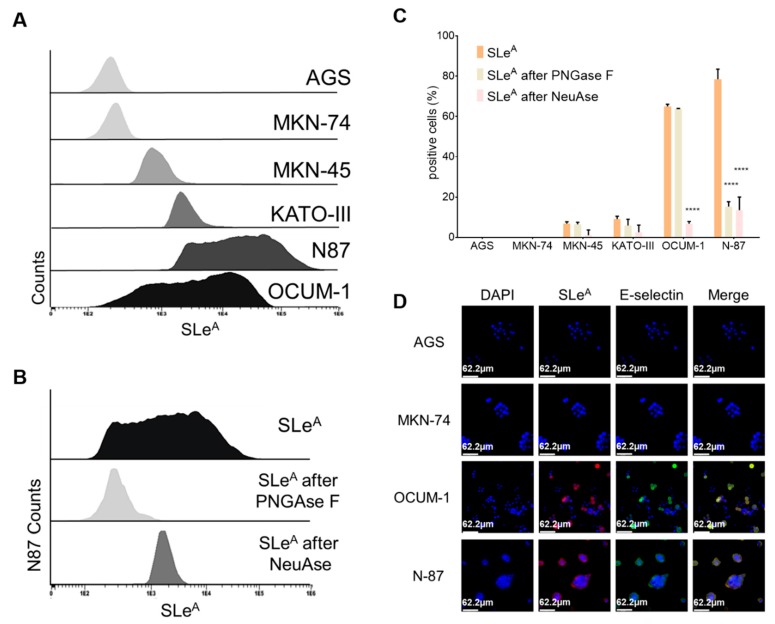
Expression of SLeA and affinity to E-selectin of different GC cell models. (**A**) SLeA expression in different cell models (AGS, MKN-74, MKN-45, KATO-III, N87 and OCUM-1). The FACS histogram in panel A highlights the levels of SLeA in AGS, MKN-74, MKN-45, KATO-III, N87 and OCUM-1 cells, ranked from the lowest to highest expression. Notably, AGS and MKN-74 were negative for the antigen, whereas N87 and OCUM-1 presented the highest expression levels. (**B**) Evaluation of the origin of SLeA expression based on N-deglycosylation with PnGase F. The histogram in panel B corresponds to control experiments involving PNGase F (for N-deglycosylation) and neuraminidase (NeuAse, sialidase) for cell line N87. This confirmed the specificity of the anti-SLeA antibody for sialylated structures and that the SLeA signals arises mostly from N-glycans. However, a still significant subpopulation of cells conserved SLeA antigen expression, suggesting also the presence of O-glycans and/or glycolipids carrying this alteration. (**C**) Summary of SLeA expression levels before and after N-deglycosylation and neuraminidase digestion for the six cell lines. Panel C summarizes SLeA evaluation for each cell line resulting from flow cytometry experiments, in agreement with the observations in panel A. Exposure to PNGase F had little effect on MKN-45, KATO-III and OCUM-1 cell lines, suggesting that the majority of the SLeA may arise from the O-glycosylation of proteins and glycolipids. In N87 the major source of SLeA expression appears to be N-glycans. Nevertheless, considerable amounts of SLeA remain, suggesting that SLeA is also carried by O-glycans. Moreover, these signals were significantly decreased when cells were treated with neuraminidase prior to analysis, reinforcing the specificity of antibody recognition. * *p* < 0.05; ** *p* < 0.01; *** *p* < 0.001; and **** *p* < 0.0001. (Student’s *t*-test for three independent replicates) (**D**) E-selectin affinity for SLeA negative gastric cancer cells lines (AGS and MKN-74) and cells expressing high amounts of the antigen (OCUM-1 and N87) by fluorescence microscopy. The panel shows that E-selectin binding (green) is dependent on SLeA expression (red). Nuclei were counterstained with DAPI (blue).

**Figure 2 cancers-12-00861-f002:**
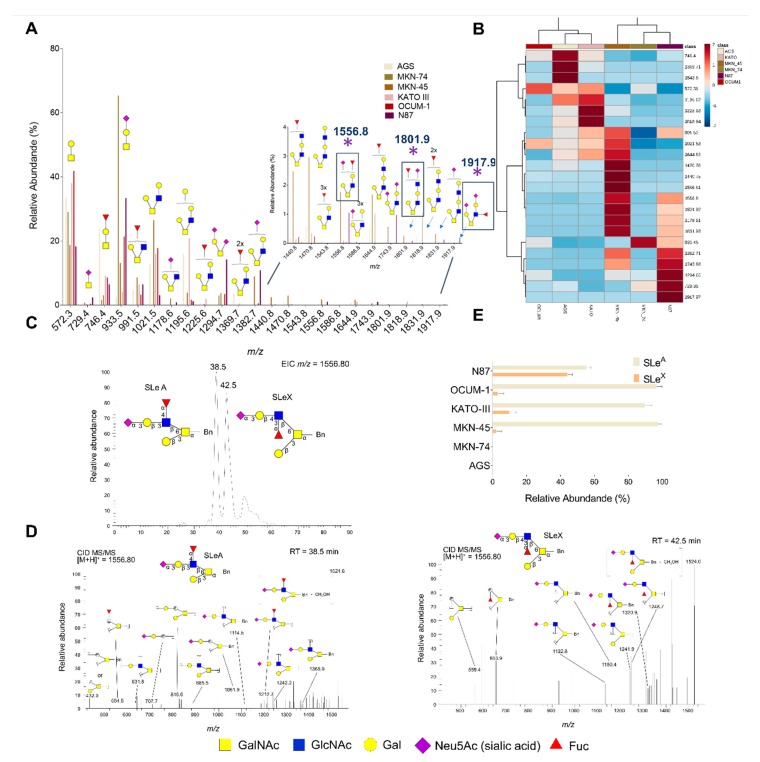
O-Glycomics analysis of GC cells. (**A**) Glycome of GC cell lines. The graph shows that the O-glycome of all cell lines was composed by fucosylated and/or sialylated core 1 and core 2 O-glycans. Additionally, shorter glycans predominated over extended structures beyond core 2. All cell lines, irrespective of their origin, overexpressed T (m/z 572.3), fucosyl-T (*m/z* 746.4), ST (*m/z* 933.5) and short core 2 (*m/z* 1021.5; 1195.6) antigens as main O-glycans. (**B**) Heat-map and cluster analysis detailing differences and similarities between the gastric cancer models. Notably, cell lines MKN-45, KATO-III, OCUM-1 and N87 also exhibited low abundant ions (<5% relative abundance; m/z 1556.8, 1801.9 and 1917.9) potentially corresponding to glycans with terminal SLeA epitopes, supporting flow cytometry analysis. (**C**) Typical nanoLC-ESI-MS/MS chromatographic profile of the N87 cell line showing the co-expression of SLeA and SLeX antigens. (**D**) Fragmentation spectra for the ions at m/z 1556 at different retention times (RT: 38.5 min for SLeA; 42.5 min for SLeX). Panel D shows one possible fragment for each main assignment, considering a mass accuracy of m/z 0.8 and glycosidic cleavages, loss of the benzyl tag and/or cross-ring fragmentations. Diagnostic signals for SLeA result from different cross-ring cleavages (0,3X, 1,3X or 3,5X) at GlcNAc residues (*m/z* = 554.5, *m/z* = 707.7, *m/z* = 815.6 and *m/z* = 1061.9). The fragment ion at *m/z* = 653.9 (Z3,5XGlcNAc), not observed for SLeA), defined SLeX Several other non-assigned fragments corresponding to multiple combinations of fragmentations (glicosydic plus cross-ring fragmentations) could also be observed. (**E**) Relative abundance of SLeA in relation to SLeX by nanoLC-ESI-MS. According to chromatographic profile showed in panel C, abundance of SLeA was estimated in comparison to SLeX, considering the ion at *m/z* 1556.8. AGS and MKN-74 did not express SLe epitopes in O-glycan chains, whereas all the other cell lines overexpressed both SLeA and, in much lower amount, SLeX.

**Figure 3 cancers-12-00861-f003:**
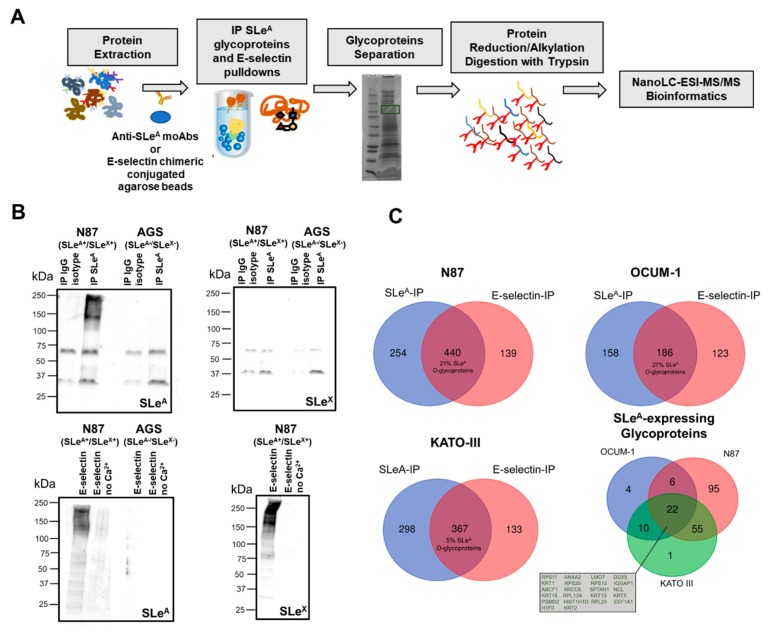
(**A**) Schematic representation of the workflow used to identify SLeA-expressing glycoproteins with affinity for E-selectin. Briefly, IPs with a monoclonal targeting the SLeA antigen were used to isolate glycoproteins from plasma membrane glycoproteins enriched extracts. In parallel, samples were pulled down with E-selectin. Parallel IPs with isotype controls and lectin pulldowns in the absence of Ca^2+^ ions (required for binding) were also performed as negative controls. The proteins were then resolved by SDS-PAGE, the bands were excised, proteins reduced, alkylated and digested with trypsin prior to analysis by nanoLC-ESI-MS/MS. (**B**) SLeA and SLeX expressions in glycoproteins obtained by IP for SLeA and corresponding isotype controls for N87 (SLeA+/SLeX+) and AGS (SLeA-/SLeX-; negative control) as well as glycoproteins with affinity for E-selectin in the presence and absence of Ca2+ (negative control). Collectively, the blots support the affinity of the IP strategy for SLeA-expressing glycoproteins with no contaminations from SLeX glycoproteins. On the other hand, E-selectin isolated both SLeA and SLeX expressing glycoproteins. (**C**) Venn diagrams highlighting the distribution SLeA-expressing glycoproteins and glycoproteins showing affinity for human E-selectin for cell line N87 (**A**), OCUM-1 (**B**), and KATO-III (**C**). Glycoproteins with confirmed O-SLeA expressing glycosites and showing affinity for E-selectin among the three cell models (**D**). The Venn diagrams in panels A–C highlight the number of glycoproteins isolated by IP for SLeA and E-selectin for each cell line, evidencing the percentage of glycoproteins commonly identified for both IPs and showing clear SLeA modifications in O-glycans. Panel C highlights the 22 glycoproteins presenting O-SLeA glycosylation and affinity for E-selectin in all cell lines.

**Figure 4 cancers-12-00861-f004:**
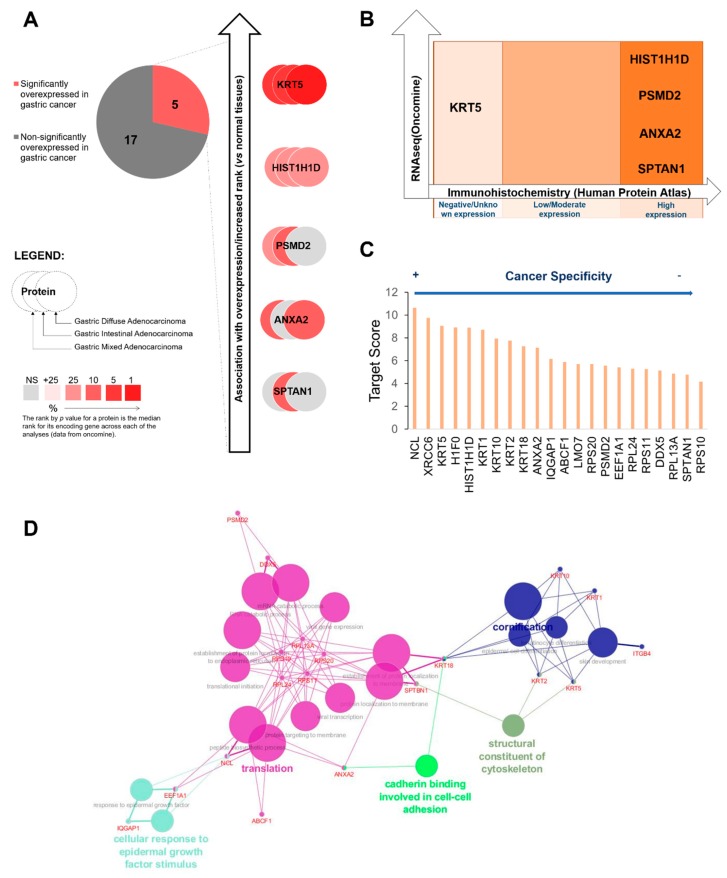
Bioinformatics analysis of the 22 glycoproteins expressing SLeA and showing affinity to E-selectin that were present in all cell lines towards biomarker discovery. (**A**) Identification of glycoproteins showing elevated gene expression. The 22 glycoproteins set was comprehensively matched against existing transcriptomics data in the Oncomine database. Only five have previously been reported to be overexpressed in GC, occurring throughout intestinal, mixed and diffuse type lesions as highlighted in the right. Notably, KRT5 and HIST1H1D were significantly overexpressed in all types, PSMD2 was mainly associated with the intestinal and mixed phenotype, ANXA2 was linked to intestinal and diffuse type but not mixed lesions, whereas SPTAN1 was overexpressed solely in mixed type lesions. (**B**) Comparison between transcripts and protein levels for the identified glycoproteins. Transcriptomics findings (RNAseq; Oncomine) were matched against protein expression in GC (immunohistochemistry, Human Protein Atlas). KRT5 showed low/no expression at the protein level, thus not reflecting increased transcriptional activity. HIST1H1D, PSMD2, ANXA2 and SPTAN1 showed both increased gene expression and high protein levels in GC. (**C**) Target score for the identified glycoproteins in GC. Briefly, higher target scores are given to glycoproteins highly expressed at the cell membrane in cancer cells and showing low levels of expression in healthy tissues. Aspects related with poor prognosis significantly contribute to increase the scoring potential. Sub-cellular re-localization of proteins from intracellular compartments in healthy cells to the cancer cell membrane, an aspect that frequently occurs in cancer, is also highly score. Conversely, proteins showing a high expression in multiple healthy tissues are penalized in comparison to those showing a more cancer-specific expression pattern. According to this protocol, NCL (that was not found overexpressed in cancer tissues) ranked first as a potentially targetable biomarker due to its cancer specific nature. (**D**) Protein–protein networks highlighting the main biological functions played by the identified glycoproteins. The map demonstrates a significant functional correlation of identified glycoproteins with cell differentiation, cytoskeleton organization, cell–cell adhesion mediated by cadherins, as well as translation and mediation of responses to epidermal growth factor stimulus. Notably, NCL appears to play a key role in the latter two processes.

**Figure 5 cancers-12-00861-f005:**
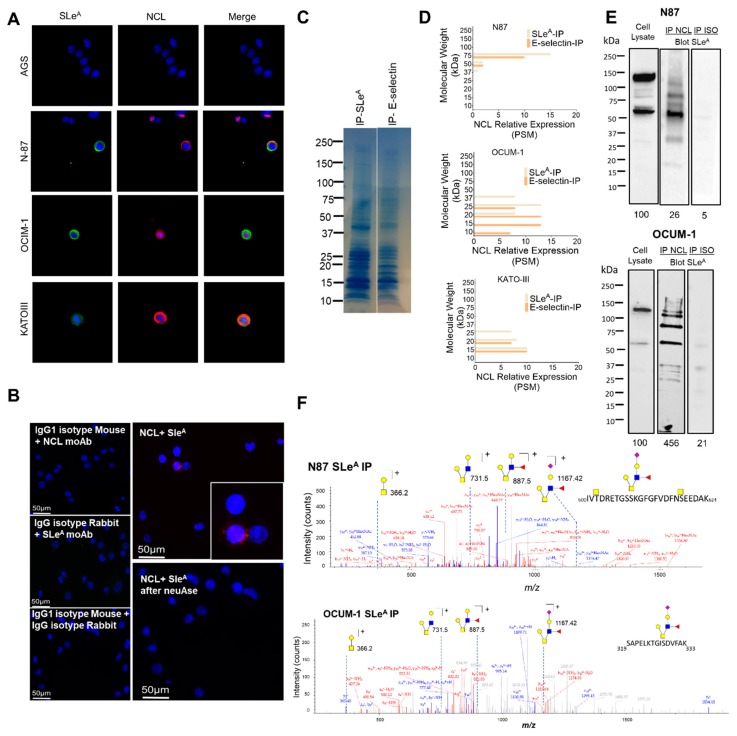
Identification of NCL-SLeA glycoforms in GC cell models. (**A**) Immunofluorescence staining of GC cells (AGS, N87, OCUM-1, KATO-III) for SLeA (green) and NCL (red). Neither SLeA nor NCL were detected in AGS cell lines. However, clear cell membrane staining was observed for both antigens in N87, OCUM-1 and KATO-III. (**B**) Proximity ligation assays for NCL-SLeA in N87 cells. Positive PLA signals (white arrows) were observed in N87 cells, which significantly decreased upon PNGase F treatment and completely disappeared in cells treated with neuraminidase. (**C**) Typical SDS-PAGE for glycoproteins isolated by IP for SLeA and E-selectin. The gels were an important starting point for molecular weight resolved identification of NCL in GC cells. (**D**) NCL expression according to molecular weight resolved glycoproteomics analysis. NCL appears between 50 and 100 kDa, with a major band just above 50 kDa. Notably, shorter proteoforms (37–15 kDa) can also be observed in OCUM-1 and KATO-III cell lines. IP-SLeA and IP-E-selectin showed similar NCL patterns. (**E**) MS/MS for NCL-SLeA glycopeptides identified by Mw-resolved glycoproteomics in N87 and OCUM-1 cell lines. Tandem mass spectra support the existence of glycopeptide fragments carrying the SLeA antigen between 50 and 75 kDa for the N87 cell line and between 25 and 37 kDa for OCUM-1 cells. (**F**) Western blots for NCL in whole cell lysates and in SLeA expressing glycoproteins isolated by immunoprecipitation (IP) for N87 and OCUM-1 cells. Neuraminidase digested samples prior to IP were used as negative controls. The blots show that SLeA IP were enriched for shorter NCL proteoforms in comparison to the whole cell extract, with emphasis on bands between 50 and 75 kDa. Bands bellow 50 kDa could also be observed in both cell lines, reinforcing MS analysis data showing low molecular weight NCL-SLeA glycoforms. SLeA IPs performed after neuraminidase digestion did not show positive NCL signals, supporting the specificity of recognition.

**Figure 6 cancers-12-00861-f006:**
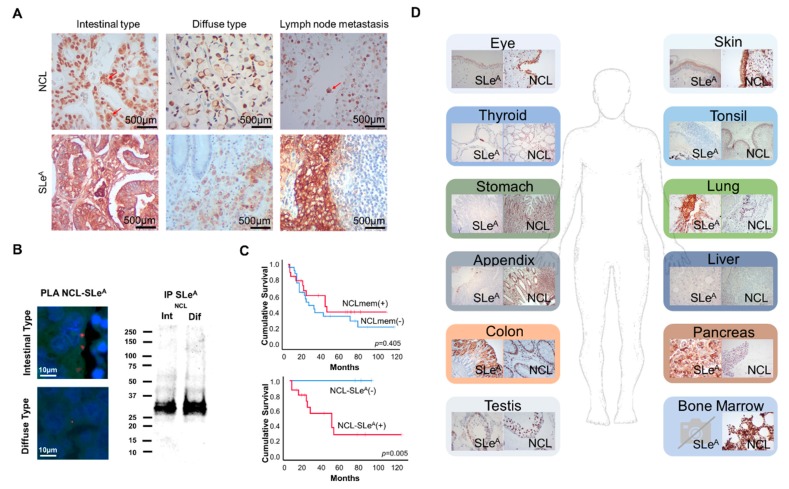
(**A**) SLeA and NCL expressions in intestinal and diffuse type GC as well as lymph node metastases (red arrows highlight NCLmem proteoforms). Both SLeA and NCL could be co-localized at the cell membrane of GC cells, reinforcing the notion of cell surface NCL-SLeA glycoforms. Of note, the nuclear subcellular location of NCL could also be observed in cancer cells. Similar observations were made in metastatic tissue. (**B**) Proximity ligation assay of SLeA and NCL (left panel) and Western blots (right panel) for NCL isolated by immunoprecipitation for SLeA. The red dots in PLA analysis suggest very close spatial proximity between NCL and SLeA. The blots from two histologically different tumors (intestinal and diffuse) show a major band at 37 kDa most likely corresponding to NCL-SLeA glycoforms. The blots also suggest the existence of vestigial proteoforms at higher molecular weights. Collectively, these findings support the existence of NCL-SLeA glycoforms at the cell surface. (**C**) Kaplan-Meier analysis of patients bearing tumors expressing NCL at the cellular membrane (NCLmem) and SLeA NCLmem (NCL-SLeA) glycoforms. The expression of NCLmem does not associate with worst prognosis (*p* = 0.405) whereas NCL-SLeA significantly associated with decreased survival (*p* = 0.05). These observations support the importance of targeting abnormal protein glycosylation towards clinically useful biomarkers. The p-values in survival analyses reflect the Breslow test. (**D**) Expression of SLeA and NCL in healthy human tissues of relevant organs. SLeA was found in the cytoplasm of exocrine pancreas cells and mildly detected at the cell membrane of goblet cells of the gastrointestinal tract, in upper layers of the corneal epithelium, rare thyroid parafollicular cells and pulmonary secretions. NCL was detected in the nucleus of most non-malignant cells. Membrane NCL was only observed in the epidermis and corneal epithelium, even though not in the same cell types expressing SLeA. These observations suggest that the conjugation of SLeA and NCL at the cell membrane is a rare event associated with cancer, holding cancer-specific targeting potential.

**Table 1 cancers-12-00861-t001:** Antibodies used in this study and corresponding experimental conditions.

Antibody	Manufacture	Reference	Clonality	Clone	Host	Application	Experimental Conditions
Anti-TPR	Novus Biologicals, Centennial, CO, USA	NB100-2866	Polyclonal	-	Rabbit	WB	1:1000, 4 h RT
Anti-ß-Actin	Sigma-Aldrich, St. Louis, MO, USA	A1978	Monoclonal	AC-15	Mouse	WB	1.4µg/mL, 1 h RT
Anti-B2M	Abcam, Cambridge, UK	ab75853	Monoclonal	EP2978Y	Rabbit	WB	1:100, 1 h, RT
Anti-CA19.9 (SLeA)	Abcam, Cambridge, UK	ab116024	Monoclonal	CA19.9-9-203	Mouse	IP, WB, IF, PLA, IHC, FC	Flow/IF—1:100, 1 h RT WB—1:1000, 1 h RT IHC—1:100, 4 °C/12 h
Anti-NCL	Abcam, Cambridge, UK	ab22758	Polyclonal	-	Rabbit	IP, WB	WB—1:1000, 1 h RT
Anti-NCL	Abcam, Cambridge, UK	ab129200	Monoclonal	EPR7952	Rabbit	PLA, IHC	1:250, 4 °C 12 h
Alexa Fluor 647 Anti-NCL	Abcam, Cambridge, UK	ab202709	Monoclonal	EPR7952	Rabbit	Flow, IF	1:100, 1 h RT
Rabbit IgG isotype	Thermo Fisher Scientific, Waltham, MA, USA	LTI-02-6102	-	-	Rabbit	IP, PLA, FC	The same as for anti-NCL
Mouse IgG1 isotype	Abcam, Cambridge, UK	ab18443	-	-	Mouse	IP, PLA, FC	The same as for anti-CA19-9
Anti-Rabbit IgG HRP	Thermo Fisher Scientific, Waltham, MA, USA	656120	Polyclonal	-	Goat	WB	1:60,000, 30 min. RT
Anti-Mouse IgG HRP	Jackson ImmunoResearch, West Grove, PA, USA	115-035-205	Polyclonal	-	Goat	WB	1:70,000, 30 min. RT
Alexa Fluor 594 Anti-Rabbit IgG	Thermo Fisher Scientific, Waltham, MA, USA	A11012	Polyclonal	-	Goat	IF	1:300, 30 min. RT
Alexa Fluor 488 Anti-Mouse IgG	Thermo Fisher Scientific, Waltham, MA, USA	A11001	Polyclonal	-	Goat	IF, FC	IF—1:300, 30 min. RT Flow—1:300, 15 min. RT
FITC Anti-human IgG	Agilent Technologies, Santa Clara, CA, USA	F0202	Polyclonal	-	Rabbit	IF	1:100, 30 min. RT
Alexa Fluor 594 Anti-mouse IgG	Thermo Fisher Scientific, Waltham, MA, USA	A11005	Polyclonal	-	Goat	IF	1:300, 30 min. RT
Recombinant Mouse E-Selectin Ig Chimera Protein	R&D Systems, Minneapolis, MN, USA	575-ES-100	-	-	Mouse	IF	1 µg/mL, 2 h RT

FC—Flow Cytometry; IF—Immunofluorescence; IHC—Immunohistochemistry; IP—Immunoprecipitation; PLA—Proximity Ligation Assay; WB—Western Blot.

**Table 2 cancers-12-00861-t002:** Clinicopathological data of patients included in the SLeA (*n* = 202) and for NCL analysis (*n* = 47).

Title	*n* for SLe^A^ (%)	*n* for NCL (%)
Stage		
I	18 (9%)	2 (4%)
II	60 (30%)	12 (26%)
III	89 (44%)	25 (53%)
IV	35 (17%)	8 (17%)
Tumor (T)		
T1	12 (6%)	0 (0%)
T2	17 (8%)	0 (0%)
T3	105 (52%)	26 (55%)
T4	64 (32%)	21 (45%)
Missing Information	4 (2%)	0 (0%)
Lymph node metastasis (N)		
N0	49 (24%)	10 (21%)
N1	40 (20%)	8 (17%)
N2	32 (16%)	7 (15%)
N3	76 (37%)	21 (45%)
Missing Information	5 (3%)	1 (2%)
Distant metastasis (M)		
M0	151 (75%)	36 (77%)
M1	36 (19%)	8 (17%)
Missing Information	15 (7%)	3 (6%)
Lauren Classification		
Intestinal subtype	124 (61%)	23 (49%)
Mixed	22 (11%)	0 (0%)
Diffuse	51 (25%)	24 (51%)
Missing Information	5 (2%)	0 (0%)
Borrmann Classification		
Type I	9 (5%)	1 (2%)
Type II	76 (38%)	18 (38%)
Type III	42 (21%)	7 (15%)
Type IV	58 (29%)	18 (38%)
Missing Information	17 (8%)	3 (6%)
Overall Survival (OS)		
Median (months±SD)	26 ± 34	28 ± 32
Mean (months±SD)	40 ± 34	40 ± 32
Age	64 ± 13	66 ± 12
Gender		
Male	109 (54%)	23 (49%)
Female	93 (46%)	24 (51%)

**Table 3 cancers-12-00861-t003:** Expression of SLeA and NCL at the cell membrane (NCLmem) in GC according to clinicopathological variables.

Clinicopathological Variables	SLeA Expression *n* (%)	*p**-Value	NCLmem *n* (%)	*p**-Value	NCLmem+SLe^A^ *n* (%)	*p**-Value
**Stage**						
I	8 (44%)	0.020	2 (100%)	0.337	0 (0%)	0.772
II	37 (62%)		4 (33%)		4 (33%)	
III	50 (57%)		12 (48%)		9 (36%)	
IV	29 (83%)		3 (38%)		3 (38%)	
**Tumor size/extension (T)**						
T1	6 (50%)	0.633	-	0.716	-	0.927
T2	11 (65%)		-		-	
T3	68 (65%)		11 (42%)		9 (35%)	
T4	36 (63%)		10 (48%)		7 (33%)	
**Lymph node metastasis (N)**						
N0	28 (57%)	0.651	4 (40%)	0.862	2 (20%)	0.551
N1	24 (60%)		3 (38%)		2 (25%)	
N2	18 (56%)		4 (57%)		3 (43%)	
N3	50 (67%)		10 (48%)		9 (43%)	
**Distant metastasis (M)**						
M0	84 (56%)	0.005	18 (50%)	0.522	13 (33%)	0.941
M1	29 (81%)		3 (38%)		3 (38%)	
**Lauren Classification**						
Intestinal subtype	77 (62%)	0.191	11 (48%)	0.671	8 (35%)	0.917
Mixed	10 (46%)		-		-	
Diffuse	35 (69%)		10 (42%)		8 (33%)	
**Borrmann Classification**						
Type I	4 (44%)	0.744	1 (100%)	0.641	1 (100%)	0.228
Type II	46 (61%)		7 (39%)		5 (28%)	
Type III	27 (64%)		3 (43%)		1 (14%)	
Type IV	35 (61%)		9 (50%)		9 (50%)	

*Chi-square test for a 95% confidence interval.

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
