# Peer review of "Nucleolin-Sle A Glycoforms as E-Selectin Ligands and Potentially Targetable Biomarkers at the Cell Surface of Gastric Cancer Cells"

_cancers, 2020, doi:10.3390/cancers12040861_

Round 1

Reviewer 1 Report

Submitted for review publication presents the occurrence on the cell surface NCL-SLeA glycoforms in gastric cancer, showing its significance as a potential biomarker for poor prognosis, focusing on their roles as a molecular signatures linked to the metastatic spread.

The study allowed for screening for SLeA expression and affinity to E-selectin among six cancer cell models using flow cytometry method. The cells glycome was studied by mass spectrometry, to validate the obtained observations, with the aim to better describe the structure of the glycans exhibiting sialylated Lewis antigens as terminal epitopes, with main focusing on the O-glycome. The analysis showed that SLeA is not a main determinant of the O-glycome but revealed that it is a leading sialylated Lewis O-glycoform among gastric cancer cells.

The authors investigated the characterization the SLeA glycoproteome among multiple gastric cancer cell lines, to discover the molecular signatures for better targeting of the cancer. They showed the occurrence of a wide spectrum of glycoproteins with increased affinity for E-selectin, including many glycoproteins with confirmed O-SLeA glycoforms. They presented 22 glycoproteins expressing SLeA, which showed affinity to E-selectin, were occurred in all described cell lines, with significance to biomarker revealing. The group showed the protein-protein networks using bioinformatical tools, to uncover glycoproteins role and important functions.

The Nucleolin expression was screened among cell lines. The Immunofluorescence staining performed by the group displayed that NCL was sprawling along the cell membrane and cytoplasm and did not occur in the cell nucleus; their nature was analysed by tandem mass spectrometry.

Appart from cellular studies, the manuscript encompasses screening of the patient group of 202 probands, 56 lymph node metastases for specific SLeA expression patterns. The study showed that SLeA higher expression was importantly linked with more advanced stages of the disease, and the occurence of distant metastases. Additionally the evaluation of NCL in a subset of gastric tumours and lymph node metastases was conducted. Subset of gastric tumors expressing NCL-SLeA glycoforms, displayed worst prognosis, which emphasize the  relevant point of targeting glycosilation for the accurate biomarker.

This manuscript is a big effort with comprehensive spectrum of  methods for discovering the  characterization and the role of NCL-SLeA glycoforms in the primary tumours and metastases, related to worst prognosis, which are not present in healthy human tissues, underlying targeted therapies as a big potential.  

Author Response

We thank the reviewer for the kind comments. The revised version presents minor spelling and grammar improvements and was also revised for typos. Amplifications for Figures 2, 5 and 6 were also provided addressing reviewers’ 2 minor comments.

Reviewer 2 Report

Sialyl-Lewis A (SLeA) is expressed in metastatic cell lines and metastases and interacts with E-selectin. The authors demonstrated that the gastric cancer cell lines N87 and OCUM-1 cells, and to a lower extent MKN-45 and KATO-III cells, expressed high levels of this glycan; AGS and MNK-74 cells did not express SLeA. Using IP for SLeA, they identified nucleolin as a major protein glycosylated in metastatic cells. Lastly, they confirmed that SLeA colocalize with nucleolin at the membrane of metastatic cell lines and that, nucleolin -SLeA glycoforms are expressed in the membrane of primary tumors and metastases of human patients with gastric cancer. These data indicate that nucleolin carrying sialylated antigens is a new potential marker of cancer metastasis.

The manuscript is very well designed and data are clear. However, the paper is too wordy and should be simplified. For example, the first paragraph on Page 12 should be shortened and these information moved to introduction and/or Methods section. Here are some minor comments:

1/ The bars on Fig. 2A are not well visible. Please, increase the size of this figure.

2/ Show higher magnification of Fig. 5AFigs. 6A, 6B.

3/ Show NCL IF in AGS cells and/or MKN-74.

Author Response

We thank the reviewer for the kind comments. The revised version presents minor spelling and grammar improvements and was also revised for typos. The first paragraph on Page 12 was moved to the methods section, in accordance with the reviewer’s comments. The revised word document is included with track changes.

General comment:

We understand that the manuscript entails a great deal of detail. This was intentional to meet the editorial criteria, which encourages scientists to publish their experimental and theoretical results in as much detail as possible. As such, we would like to keep it in its present form.

Minor comments:

Comment #1:  The bars on Fig. 2A are not well visible. Please, increase the size of this figure.

Answer #1: We have increased the size of Fig. 2A. However, its final form and zoom on the manuscript depends on the editorial office.

Comment #2: Show higher magnification of Fig. 5AFigs. 6A, 6B.

Answer #2: We have done as requested.

Comment #3: Show NCL IF in AGS cells and/or MKN-74.

Answer #3: This data was presented in Supplementary Figure S3
